# Chronic Pain Prevalence and Psychosocial Burden in the Italian Population from the 2019 European Health Interview Survey

**DOI:** 10.3390/ijerph22091395

**Published:** 2025-09-06

**Authors:** Alice Maraschini, Michael Tenti, William Raffaeli, Laura Iannucci, Lidia Gargiulo, Alessandra Burgio, Giada Minelli, Corrado Fagnani, Emanuela Medda, Maurizio Ferri, Miriam Salemi, Virgilia Toccaceli

**Affiliations:** 1Istituto Superiore di Sanità, Statistical Service, 00161 Rome, Italy; alice.maraschini@iss.it (A.M.); giada.minelli@iss.it (G.M.); 2Fondazione ISAL, Institute for Research on Pain, 47921 Rimini, Italy; william.raffaeli@fondazioneisal.it; 3ISTAT—National Institute of Statistics, 00184 Rome, Italy; iannucci@istat.it (L.I.); gargiulo@istat.it (L.G.); burgio@istat.it (A.B.); 4Istituto Superiore di Sanità, Centre for Behavioural Sciences and Mental Health, 00161 Rome, Italy; corrado.fagnani@iss.it (C.F.); emanuela.medda@iss.it (E.M.); maurizio.ferri@iss.it (M.F.); miriam.salemi@iss.it (M.S.); virgilia.toccaceli@iss.it (V.T.)

**Keywords:** chronic pain, epidemiology, European Health Interview Survey, prevalence, Italy, psychosocial burden, depression, anxiety, activity limitations, social participation, work, treatment access, health disparities

## Abstract

Chronic pain (CP) is a global healthcare concern requiring careful monitoring. In Italy, the most recent CP prevalence estimates date back to 2003. In this work, we analyzed data from the 2019 European Health Interview Survey, based on a representative sample of the Italian population (n = 44,492), to update national CP prevalence estimates and evaluate its psychosocial burden. CP was detected using a validated questionnaire. Our results show that approximately 10.5 million adults (24.1%) suffer from CP; prevalence and intensity are higher among females and increase with age. In 54.3% of cases, CP was triggered by a diagnosed disease, while 13.6% remain undiagnosed and 14.8% do not seek treatment for CP. Severe CP accounts for 29.4% of cases. Individuals with severe CP are significantly more likely to experience difficulties in social participation (OR 4.82; CI 4.41–5.27), increased work absences (OR 4.18; CI 3.53–4.94), depression (OR 7.10; CI 6.22–8.11), and greater use of primary (OR 2.90; CI 2.64–3.18) and specialist healthcare (OR 2.63; CI 2.40–2.89) as well as diagnostic procedures (OR 2.27; CI 2.07–2.49). Among subjects diagnosed with depression or severe chronic anxiety, CP seems to reduce access to mental health care (OR 0.75; CI 0.61–0.92) and increase abandonment due to financial barriers (OR 1.57; CI 1.07–2.31). Unlike a few countries (e.g., Spain and Denmark) that have recorded a generally increasing trend in CP prevalence, our figures confirm a quite stable national epidemiological pattern. Our estimates underscore the need for careful management of CP and its psychosocial burden. Since data were collected just before the COVID-19 pandemic, they may represent a crucial baseline for monitoring post-pandemic trends.

## 1. Introduction

Chronic pain (CP) represents a significant public health concern worldwide that affects about 20% of people in Western countries [1]. As suggested by several experts [2], the high prevalence of CP and its substantial impact on health and well-being require adequate monitoring with validated tools to overcome methodological discrepancies and allow for comparisons of estimates among and within countries. In fact, epidemiological studies, to date, have exhibited significant methodological variability in CP assessment, regarding, for instance, the recall period (e.g., 6 vs. 12 months) or CP definition (e.g., using the adjective “persistent” vs. “continuous” and “intermittent”) and duration (e.g., lasting for at least 3 vs. 6 months) [3]. In the United States, CP prevalence was estimated at 19% in the 2010 National Health Interview Survey [4], which recorded a slight increase of about 1.5% just a few years later [5,6]; similarly, in Europe, since the results of the large survey by Breivik and colleagues in 2003 [7], many countries have updated national CP prevalence, often recording different estimates in subsequent surveys [8,9,10,11]. In Italy, the most recent prevalence estimates derive from a few surveys on limited samples or specific pain subtypes [12,13,14,15]. From a public health perspective, therefore, a need has emerged to update national estimates focusing on CP as a whole phenotype (i.e., irrespective of different conditions or diagnoses), considering that CP outcomes and burden depend more on the severity and interference of pain than on the underlying causes [3].

Furthermore, no previous studies in Italy have investigated the psychosocial burden of CP at the general population level. Indeed, syndromes such as back pain and migraine are among the main causes of years lived with disability [16,17], contributing to sleep disturbances, anxiety, depression, and an overall decrease in physical and psychological health [18]. These consequences and comorbidities may, in turn, affect CP and its management: depression, for instance, has been linked to increased pain and functional limitations, greater use of medical services, and less favorable treatment outcomes [19]. Moreover, CP significantly affects family environment, social interactions, and work ability [18], while placing a substantial economic burden on society through its direct and indirect costs [20]. Despite this impact, CP remains often inadequately managed, and affected subjects are less likely to access mental health services than those who are not affected, leading to unnecessary suffering, diminished quality of life, and higher social and healthcare costs [21,22].

In this framework, the objectives of the present analysis were (A) to update CP prevalence estimates in Italy using a validated questionnaire, exploring pain intensity, possible triggers/causes, treatment modalities, and perceived effectiveness; (B) to explore the burden on psychosocial well-being and daily life activities among the general population; and (C) to examine whether, among people with diagnosed depression and/or severe chronic anxiety, CP might influence (i) the propensity to consult a psychologist, psychotherapist, or psychiatrist, (ii) the likelihood of forgoing mental health care due to financial barriers, and (iii) the propensity to consider psychological/psychiatric therapy a need.

From a preventive perspective, updated epidemiological patterns of CP and its psychosocial burden might help identify at-risk subgroups, prompting international comparisons and collaboration to facilitate the development of targeted prevention initiatives.

## 2. Materials and Methods

The present work analyzed the cross-sectional data collected by the Italian National Institute of Statistics (ISTAT) in 2019 on a representative sample of the Italian general population, as part of the 2019 European Health Interview Survey (EHIS), wave 3 [23]. While the overall EHIS target population comprised individuals aged 15 and above, the present analysis, for objectives A and B, focused on EHIS adult respondents (>18 years), owing to the very low numbers of CP cases among younger respondents (<18 years). For objective C, we used the EHIS subpopulation of adults with diagnosed depression and/or severe chronic anxiety. To carry out the analyses, an interdisciplinary Working Group (GDL-DC) was established under the coordination of the Istituto Superiore di Sanità (ISS), with contributions from experts of the ISAL Foundation (Institute for Pain Research) and the Italian National Institute of Statistics (ISTAT).

### 2.1. EHIS National Population Sampling and Data Collection

The EHIS sample design was carried out in accordance with the guidelines in Eurostat’s methodological manual [23]. The sampling strategy followed a two-stage (municipalities–households) stratified design; more information is available at https://ec.europa.eu/eurostat/web/products-manuals-and-guidelines/w/ks-gq-24-018 (accessed on 7 August 2025). The theoretical final sample included 837 municipalities and 30,142 households and was representative of the Italian resident population. The achieved sample size was approximately 22,800 households and 52,442 individuals. All household members belonging to “de facto” families were interviewed.

Data collection was carried out in 2019 in two waves: April–June and September–December. The questionnaire modules were those established by the European Regulation; however, Italy added supplementary forms to address national information needs, including the CP questionnaire. Data were collected using the paper-and-pencil technique [24], through face-to-face, door-to-door interviews conducted by trained interviewers. However, the CP questionnaire was included in the self-administered section of the EHIS, together with the variables used to construct the indicator “Difficulties in social participation”, which was analyzed in this study (see the section “Other EHIS measures and indicators used to describe CP”).

### 2.2. The CP Questionnaire

The “Brief Five-Item Chronic Pain Questionnaire” (CP-QUEST) was used to assess CP [25]. It is a brief self-administered five-item questionnaire that was validated by our group and specifically developed to detect CP and some pain-related characteristics in the general population. The first item aims to detect CP and is based on the IASP definition [2]. It asks respondents whether, at the time of the survey, they have been suffering from persistent physical pain for more than three months. Following this filter question, four close-ended items assess (i) pain intensity, using a five-level categorical verbal scale, (ii) possible underlying (group of) causes or triggers of CP, (iii) frequency of current or past treatments (indistinctly, drugs or other therapies), and (iv) self-perceived effectiveness of treatments. The CP-QUEST showed a good understandability, adequate test–retest reliability, and good construct and criterion validity [25].

### 2.3. Other EHIS Measures and Indicators Used to Describe CP

#### 2.3.1. Limitation in Activities Due to Health Problems

The indicator measures the presence of long-term limitations (i.e., lasting at least six months) in performing daily activities (e.g., working, studying, and leisure activities) due to health problems. Three levels of limitation are assessed: “No limitation”; “Non-severe limitation”; and “Severe limitation”. This indicator is derived from the routed version of the Global Activities Limitation Indicator (GALI) [26] and uses the following questions: “Are you limited because of a health problem in activities people usually do?” and “Would you say you are: severely limited, limited but not severely, or not limited at all?”. If the answer is “Severely limited” or “Limited but not severely”, the subsequent question is “Have you been limited for at least the past six months?” (responses: yes or no). The variable was dichotomized into “No limitations” and “With limitations” (i.e., severe or non-severe).

#### 2.3.2. Difficulties in Social Participation

The indicator detects difficulties in participating in usual social activities, such as going out with family and friends, going to the cinema or theater, attending music events, festivals, or fairs, and visiting art galleries, museums, or other places of cultural interest. The answer options are “No difficulties”, “Some difficulties”, “A lot of difficulties”, “I can’t do it at all”, and “I’m not interested in the activity”. The response categories “I’m not interested in the activity” and “No difficulties” were recoded into “No difficulties or interest”, while “Some difficulties”, “A lot of difficulties”, and “I can’t do it at all” were recoded into “Presence of difficulties”.

#### 2.3.3. Absence from Work Due to Personal Health Problems

This indicator measures, for employed respondents, the number of days of absence from work due to health reasons over a 12-month period. The variable was recoded into two categories: “No absence or up to three days of absence” and “Four or more days of absence”. In Italy, health-related absences from work up to three days generally do not require medical certification, as they are usually due to non-serious health problems (e.g., seasonal illnesses).

#### 2.3.4. Depression

This question refers to subjects reporting physician-diagnosed depression in the past 12 months.

#### 2.3.5. Severe Chronic Anxiety

This question refers to subjects reporting physician-diagnosed severe chronic anxiety in the past 12 months.

#### 2.3.6. Primary Care Consultations

Participants reporting a primary care consultation within the previous six months were asked how many times they had consulted their general practitioner in the past four weeks. The response options were “Number of primary care physician consultations in the past four weeks” and “I have not consulted my primary care physician in the past four weeks”. The variable was recoded into two categories: “No primary care consultations in the past four weeks” and “One or more primary care consultations in the past four weeks”.

#### 2.3.7. Specialist Medical Consultations

Participants reporting a specialist medical consultation (e.g., ophthalmology, orthopedics, cardiology, gynecology, or psychiatry) within the previous six months were asked how many times they had consulted a specialist in the past 4 weeks. The response options were “Number of specialist medical consultations in the past four weeks” and “I have not consulted a specialist in the past four weeks”. The variable was recoded into two categories: “No specialist medical consultations in the past four weeks” and “One or more specialist medical consultations in the past four weeks”.

#### 2.3.8. Diagnostic Procedures

Participants were asked whether they had undergone any diagnostic procedures in the past 12 months. The question was “In the past 12 months, have you undergone any specialized medical tests, such as X-rays, ultrasounds, MRI, CT scans, mammograms, Doppler ultrasounds, echocardiograms, electrocardiograms, Pap smears, or other diagnostic procedures? Please exclude any tests performed during hospital admissions or day hospital visits”. Response categories were “Yes” and “No”.

#### 2.3.9. Consultation of a Psychologist, Psychotherapist, or Psychiatrist

The question referred to outpatient visits with a psychologist, a psychotherapist, or a psychiatrist in the past 12 months. Answer categories were “Yes” and “No”.

#### 2.3.10. Foregone Mental Health Care Due to Financial Barriers

This indicator assesses difficulty accessing mental health services due to financial reasons. The question was as follows: “Was there any time in the past 12 months when you needed mental health care from a psychologist, psychotherapist or a psychiatrist, but you could not afford it?” Response categories were “Yes”, “No”, and “No need for mental health care”. Both medical (psychiatrists) and non-medical (psychologists) care are included.

#### 2.3.11. Multimorbidity

This indicator identifies subjects suffering from three or more chronic diseases among the following chronic conditions investigated within the EHIS: asthma; chronic bronchitis, Chronic Obstructive Pulmonary Disease; emphysema; myocardial infarction or chronic consequences of myocardial infarction; coronary heart disease or angina pectoris; hypertension; other heart diseases; stroke or chronic consequences of stroke; osteoarthritis; lower back disease or other chronic back condition; cervical disease or other chronic neck condition; diabetes; allergy; liver cirrhosis; urinary incontinence, bladder control issues; kidney problems; chronic kidney failure; depression; severe chronic anxiety; malignant tumor; Alzheimer’s disease; senile dementias; Parkinsonism; and other chronic diseases.

### 2.4. Personal Data Treatment

This CP survey, conducted within the framework of the EHIS, was included in the Italian “National Statistical Programme” (NSP) that encompasses all the statistics considered of high public relevance in Italy. The NSP provides the juridical bases for the collection of personal data and health-related data in particular. Within the NSP, the Italian National Authority for the Protection of Personal Data exerted—according to the principles of accountability and privacy by design—an ex ante control on the compliance with personal data collection legal requirements (GDPR 679/2016 and the Italian Legislative Decree n. 196, 30 June 2003, and subsequent modifications). Moreover, the National Authority for the Protection of Personal Data established that there was no obligation to respond to the CP module as well as to other EHIS modules addressing health issues.

### 2.5. Data Analyses

The EHIS estimates were reported to the Italian general population in terms of absolute values and percentages; they were calculated using a constrained weighting estimator. Each survey unit was assigned a weight reflecting the number of units in the population represented by that unit. Full details of the methodology are provided in the Eurostat quality report [27].

Socio-demographic, clinical, and psychosocial characteristics of participants were stratified by CP status and presented as sample numbers (n), weighted proportions (%), and corresponding 95% confidence intervals (95% CIs). All the percentages were reported excluding missing data. Comparisons between respondents with and without CP for the variables included in Table 1 were performed using the chi-square test.

Statistical analyses included several categorical variables. Age was categorized into four classes: 18–34; 35–64; 65–74; and 75+. Education level was classified into three categories: “Low” (no formal education up to middle school diploma); “Medium” (high school diploma); and “High” (three-year university degree or higher). In addition, responses regarding “possible causes of chronic pain”, “treatment frequency,” and “perceived effectiveness” were analyzed by pain intensity and grouped into three categories: mild (“very mild” and “mild” combined); moderate; and severe (“severe” and “very severe” combined).

CP prevalence was estimated by sex and age class. Age- and sex-standardized rates were calculated for geographical macro-areas (Northern Italy, Central Italy, and Southern Italy and Islands) using the direct method, with the 2013 European Standard Population as the reference population.

### 2.6. Multivariate Analyses

Logistic regression models were fitted to examine the association of CP (exposure variable) with various indicators of psychosocial impact and healthcare utilization (outcome variables), while taking into account the effect of age, sex, education level, and, in one model, multimorbidity (confounding variables). Covariates were selected based on their established relevance in the literature, their potential to confound the relationship between CP and the outcomes under study, and the differences observed between groups as reported in Table 1.

To assess the impact of CP and its intensity on psychosocial conditions, the following outcomes were evaluated: (i) limitations in activities due to health problems, (ii) difficulties in social participation, (iii) absence from work due to personal health problems (among employed participants), (iv) self-reported diagnosed depression, and (v) severe chronic anxiety. CP intensity was included in the model as the main risk factor, using “absence of CP” as the reference category. Age class, sex, and education level were included as confounders in the models to adjust the estimates. Crude and adjusted odds ratios, along with their 95% CIs, were reported.

Similarly, the impact of CP and its intensity on access to primary and specialist medical consultations (in the 4 weeks prior to the survey) and on diagnostic procedures (in the 12 months prior to the survey) was evaluated, including “multimorbidity” as a covariate in the model, to account for the burden of coexisting chronic conditions that may influence healthcare utilization, independently of CP.

All logistic regression models were applied to the whole sample and were calibrated with sample weights.

Finally, the potential influence of CP on the propensity to access mental health services was assessed among EHIS respondents reporting a diagnosis of depression and/or severe chronic anxiety in the past 12 months. To this end, the main socio-demographic characteristics of this subpopulation were firstly described as being stratified by CP status. Three logistic regression models were then applied. Due to the small sample size of this subpopulation (n = 2411), only CP occurrence was analyzed to avoid the likely poor reliability of using CP intensity levels.

Model fit was assessed using the Area Under the Curve (AUC) statistic, which indicates how well the model discriminates between the two outcome classes; values closer to 1 indicate better discriminative performance of the model.

Data were analyzed using SAS (version 9.4, © 2023 SAS Institute Inc., Cary, NC, USA).

## 3. Results

### 3.1. CP Prevalence: Sex, Age, Geographical Distribution, and Associated Characteristics

Respondents aged 18 years and older who completed the CP-QUEST comprised 38,775 out of a total of 44,492 EHIS participants, giving a response rate of 87%. Respondents and non-respondents did not differ for the main socio-demographic characteristics, taking into account missing values (see Appendix A).

Table 1 reports participants’ characteristics stratified by CP status. Among subjects with CP, there is a higher prevalence of women (60.7% vs. 39.3%), people over 65 years of age (48.1% vs. 20.5%), and subjects with a low level of education (60.9% vs. 39.1%). All psychosocial indicators, as well as multimorbidity (at least three comorbidities), show consistently higher percentages among participants with CP compared to those without CP. These differences were statistically significant (*p* < 0.001) when comparing CP absence with both CP presence.

Out of the total adult population residing in Italy, 24.1% declared to suffer from a persistent pain in one or more parts of the body that had been lasting, at the moment of the survey, for more than 3 months. This prevalence accounts for approximately 10.5 (95% CI 10.3–10.8) million people and might even be underestimated due to the exclusion of missing responses to the whole EHIS (13% out of the total eligible sample). In absolute figures, this underestimation might result in an approximate increase of up to 1 million people affected by CP.

Prevalence shows an increasing trend by age, going from 8% among the younger respondents (18–34 years of age) to over 50% among people aged 75 years and older. Sex differences with unfavorable rates for females begin to emerge at 35 years of age, becoming larger for older age groups (see Table 2).

As regards geographical macro-areas, sex- and age-standardized prevalence rates provide a homogeneous picture of CP distribution over the whole national territory: 22.9% in the North, 20.9% in the Center, and 22.7% in the South and Islands. However, among people older than 65 years, a geographical gradient can be detected, with the South and Islands showing the highest rates (39.1% in the North, 39.5% in the Center, and 46.1% in the South and Islands). Moreover, female disadvantage is evident in all the macro-areas, with the largest sex gap in the South and Islands.

Figure 1 and Figure 2 illustrate CP characteristics by age and sex (the same data are also provided as a table in Appendix A). As concerns CP intensity levels, the majority of CP-affected subjects suffer from moderate pain (52.4%). Severe pain affects 29.4% of CP-affected subjects, with a higher percentage among females (32.4% vs. 24.8%), showing an increasing trend by age.

Among the whole CP population, the most frequently reported underlying CP cause/trigger is a “disease diagnosed by a physician” (54.3%), especially among the older groups. In contrast, CP following a trauma event is the most frequently reported cause (46.7%) among the 18–34 age group. Individuals who reported not yet having received a specific diagnosis account for 13.6%, and this figure increases up to 21% among younger respondents (18–34 years).

As regards CP treatments, 14.8% of the respondents declared they were not assuming any medication or undergoing any treatment, while 36.6% treated CP when needed, and 32.3% continuously treated it. Furthermore, among those treating CP, whatever the modality of treatment, 69.1% experienced partial relief, while 7.6% reported no relief.

Figure 3 shows the CP intensity distribution in relation to the underlying CP causes or triggers. Cancer shows the highest frequency of severe/very severe CP (40.9%). This percentage decreases to 23.9% among those who do not yet know the cause or the trigger of the onset and persistence of their CP.

Among CP-affected subjects who declared that they treat their pain continuously, 47.4% suffer from severe or very severe CP. This percentage decreases to 30.3% and 20.7%, respectively, among those who undergo cycles of therapy and those who treat CP when needed. Of note, among those who do not assume any medications or undergo any treatments, the proportion of those with severe CP is lower but not negligible, accounting for 11.7%.

Among CP sufferers who declared the effectiveness of the therapy, 18.2% suffer from severe CP, while among those who consider the therapy ineffective, the percentage of severe CP rises to 55.3%.

### 3.2. CP Psychosocial Burden and Access to Health Services Among CP-Affected Subjects

CP-affected subjects show a higher prevalence of mental health disorders such as depression (11.3% vs. 2.1% in those without CP) and severe chronic anxiety (8.1% vs. 1.5% in those without CP) (Table 1). Table 3 and Table 4 report a multivariate analysis of the psychosocial burden of CP. The risk of impaired daily activities among CP sufferers, compared to those unaffected, is threefold higher when CP is mild in intensity (ORadj: 3.02), sixfold higher when CP is of moderate intensity (ORadj: 6.25), and over fifteenfold higher when CP intensity is severe or very severe (ORadj: 15.99). Severe or very severe CP is associated with an increased risk of having difficulties in social participation (ORadj: 4.82) and of being absent from work for more than three days per year (ORadj: 4.18). In Table 4, the adjusted ORs indicate an approximately sevenfold increase in both diagnosed depression (ORadj: 7.10) and severe chronic anxiety (ORadj: 7.03) among subjects with severe or very severe CP compared to unaffected individuals.

As shown in Table 5, as pain intensity increases, the risk of primary care or specialist medical consultations in the 4 weeks prior to the interview also increases, and similarly for the risk of diagnostic procedures in the past 12 months. These risks are more than twofold higher when CP is severe or very severe (ORadj: 2.90 for primary care consultations; 2.63 for specialist medical consultations; 2.27 for diagnostic procedures).

### 3.3. Prevalence of CP and Use of Mental Health Service in the Whole EHIS Sub-Population Diagnosed with Depression and/or Severe Chronic Anxiety

In the adult population surveyed in the EHIS, 5.4% of the respondents (about 2.7 million individuals) reported having received a diagnosis of depression (4.2%) and/or severe chronic anxiety (2.9%) within the past 12 months. CP prevalence in this subpopulation was 61% (95% CI 58.6–63.4%), with sex-specific estimates of 52.5% (95% CI 47.8–57.3%) in males and 64.5% (95% CI 61.7–67.3%) in females. Sex differences were more pronounced in the 35–64 and 65–74 age groups (44.7% male vs. 56.9% female and 52.1% male vs. 65.9% female, respectively).

Table 6 reports the socio-demographic characteristics of this subpopulation, according to CP status. Women are more represented overall (70.6%), particularly among those who suffer from CP (74.7%); furthermore, individuals with CP were more frequently aged 75 years or older.

The main reason for focusing on this subpopulation was to compare subjects with and without CP in terms of consultation with mental health professionals and possible reasons for foregoing mental health services.

Among respondents in the EHIS subpopulation who reported a diagnosis of depression and/or severe chronic anxiety, the rates of consultation with a psychiatrist/psychologist/psychotherapist in the 12 months prior to the interview were 22.2% and 33.9% in subjects with and without CP, respectively; this difference was statistically significant (*p* < 0.001). Furthermore, 6.9% of CP-affected subjects and 5.2% of CP-unaffected patients had discontinued mental health services in the previous 12 months for financial reasons, a difference that was also significant (*p* = 0.018). The logistic regression model (Table 7), adjusted for sex, age, and education level, confirmed that in the population suffering from diagnosed depression or severe chronic anxiety, people with CP might have a lower likelihood of consulting a psychiatrist, psychologist, or psychotherapist compared to CP-unaffected subjects (OR = 0.75; 95% CI: 0.61–0.92). Unmet needs for mental health care due to financial barriers were more likely among subjects with CP (OR = 1.57; 95% CI: 1.07–2.31), whereas the belief of not needing mental health care did not differ between those with and without CP (OR = 1.02; 95% CI: 0.85–1.23).

## 4. Discussion

The present work reports a 24% prevalence of CP in a broad and representative sample of the general Italian population, accounting for more than 10 million people in the country. This prevalence estimate is slightly lower than the estimate reported by Breivik and colleagues (26%) for Italy in 2003 [7] and comparable to the CP prevalence detected in 2014 in a sample of the general adult population residing in Narni (Umbria, Italy) (28.4%) [14] as well as that recently observed in another Italian survey conducted on a sample of twins (24%) [15]. Consistent with previous studies [11,28,29,30,31,32,33,34,35,36,37,38], we observed higher CP estimates and CP intensity levels among females. These sex-related disproportions might have multifactorial origins. Among biological factors, distinct pain circuitry, endogenous pain modulatory systems, immune mechanisms (such as variations in microglial activation), and hormonal and genetic factors might significantly contribute to sex differences in nociception and analgesia. Psychosocial factors such as anxiety, depression, and gender stereotypes also play a role [36,37,39]. For both sexes, the prevalence increases with age. This age-related trend may be due to neurobiological changes in pain perception pathways. Furthermore, it may be explained by the higher comorbidity observed in the elderly between CP and pathologies such as osteoarthritis and diabetes, which, in advanced stages, are characterized by persistent pain [6,40,41].

Unlike a few countries, such as Spain and Denmark, which show a generally increasing trend in CP prevalence [10,11], our figures indicate a relatively stable epidemiological pattern. Moreover, our data show a geographic gradient in the older population (65 years and over), with unfavorable estimates for Southern Italy, which is opposite to the pattern previously detected by Breivik and colleagues [7]. The higher prevalence of CP among older subjects in Southern Italy may be attributed to various factors. Among these, socioeconomic disparities, including lower income levels and educational attainment, reduced awareness, and a lack of health literacy, may play a key role [42]. All of these factors represent a significant risk for CP [43] and other comorbidities that affect older people in Southern Italy more frequently [44]. These comorbidities, in turn, represent a further risk factor for CP [45]. This situation may be exacerbated by the greater difficulty in accessing healthcare services that characterizes the southern regions of Italy [46].

The Italian CP estimates are higher than those reported in the U.S. in 2016 and 2019 (approximately 20%) [6] and lower than those found in Spain in 2022 (25.9%) [11], Denmark (27.8%, people aged 16 years and older) [10], France (31.7%) [8], Germany (32.9%, people aged 14 years and older) [9], and Portugal (36.7%) [33]. These differences, however, cannot be thoroughly evaluated, as they may be influenced by methodological reasons (e.g., sampling strategies, measures, and CP definitions) [3] and several other factors (e.g., differences in lifestyles, age stratification, and treatment choices) [7].

It is noteworthy that almost one-third of the respondents (i.e., more than 3 million people) reported a “severe” or “very severe” CP intensity, showing a substantial physical and psychosocial impact. CP-affected subjects, compared to non-affected subjects, face substantially greater risks—ranging from 1.5-fold to even more than 14-fold higher—of experiencing limitations in daily activities and in social participation, needing longer sick leaves, and being diagnosed with depression or severe chronic anxiety. Consistent with previous findings [11,47,48], each of these risks appears to increase with the intensity of pain. With regard to emotional well-being, our results are in line with a recent study showing that U.S. adults who experience CP are about five times more likely to suffer from significant, unresolved anxiety or depression, compared to those without CP [22]. At the same time, CP-associated psychosocial burden may have a negative influence on CP management [20]. Comorbid depression, for instance, may worsen pain prognosis and treatment outcome and increase the direct costs that several healthcare systems have to bear [18,19,49]. Therefore, a few authors have emphasized the need for a multidisciplinary approach to manage CP patients’ conditions [18,21].

Among subjects diagnosed with anxiety and/or depression, our findings showed that CP sufferers may have a lower propensity to consult mental health services compared to CP-free subjects. Consistent with these findings, in the U.S., it was recently observed that adults with CP and mental health problems access mental health services less frequently than people with mental health problems only [50]. Several potential factors contributing to this treatment gap may be proposed, including limited awareness of the rationale for mental health treatment when CP is present, fear of stigma, or limited accessibility due to financial constraints or high costs. Further contributing factors may include practical barriers related to physical disability, fragmentation of care between pain management and mental health services, a lack of integrated treatment approaches, and insufficient training for providers to manage comorbid mental health conditions [50]. Additionally, some patients may prioritize physical pain symptoms over mental health concerns due to the urgent and debilitating nature of CP or may prefer self-management strategies over formal mental health interventions. Other patients might experience a lack of effectiveness of mental health treatments for their difficulties in adapting to the pain condition.

Regarding financial barriers, our findings suggest that when depression and/or chronic anxiety are diagnosed, economic conditions might be stronger determinants of mental health service use among CP sufferers compared to CP-unaffected people. However, CP-affected and -unaffected individuals are equally likely to believe they do not need mental health services despite these diagnoses. Consequently, our findings strengthen the notion that, in addition to specific CP barriers, more general barriers, such as societal stereotypes or fear of stigma, can influence perceptions of the need for mental health treatment.

We found that subjects with CP are more likely to use primary and specialist health care, as well as diagnostic procedures, compared to CP-unaffected subjects, and this propensity increases with pain severity regardless of other comorbidities. These findings are consistent with previous studies based on health administrative data [51] or interviews [52,53]. Furthermore, they align with a recent study showing that, among subjects with chronic conditions such as cancer or stroke, those experiencing pain are at greater risk of accessing healthcare services [54]. Further research is necessary to better elucidate these relationships.

Moreover, our work provides data on possible underlying causes or triggers of CP. The majority of the affected subjects reported CP onset following a diagnosed disease (rheumatism, arthritis, and infections are a few examples provided by the specific item). Traumatic events were reported as CP triggers in almost 22% of cases, a higher frequency than that reported in other Western countries such as Portugal (12.6%) [33] and the U.S. (9%) [55]. Conversely, the prevalence of post-surgical CP (7%) is lower than that detected in Norway (18.3%) [56] and aligns with figures from Portugal [33]. However, caution is warranted in interpreting these differences given the limited data available from general population studies on post-surgical and post-traumatic CP prevalence.

Cancer is reported as an underlying cause of pain in about 3% of cases, and cancer-related CP shows the highest intensity compared to all the other causes. These represent the most recent figures regarding cancer-related CP prevalence in the general Italian population. However, a discrepancy emerges when comparing cancer prevalence as recently assessed by the Italian Cancer Registry data (5.9%) [57] and cancer prevalence resulting from the 2019 EHIS survey (2.5%); this difference might reflect the challenges of capturing advanced-stage oncological patients by a general population survey such as the EHIS. Further insights into cancer-related CP in Italy may be found elsewhere [58].

In more than 13% of cases, CP was reported to begin after an as-yet undiagnosed disease. A recent epidemiological study in Spain found even higher proportions of CP-affected subjects who did not know the cause of their pain (27.1%) [11]. This phenomenon likely reflects complex clinical and psychosocial roots. It may include patients experiencing delays in diagnostic processes or presenting with complex symptoms and comorbidities, such as psychiatric comorbidities [59,60,61], which makes it difficult to attribute a certain diagnosis. These delays and difficulties are particularly pronounced given the lack of reliable biomarkers for conditions such as fibromyalgia or chronic pelvic pain. Moreover, this subgroup might include subjects who do not seek medical attention for their pain. Ongoing refinement of our questionnaire may help to disentangle the characteristics of CP-affected subjects with respect to underlying causes.

As regards treatments, although the great majority of respondents (almost 70%) reported continuous or as-needed CP treatments, most reported only partial or no relief. This highlights potential gaps in treatment effectiveness and underscores the need for more accurate diagnostic assessments and systematic evaluation of treatment outcomes. It is worth noting that, excluding cases of cancer-related pain, approximately 1.5 million people in Italy do not undergo any treatments for their pain, and more than 10% of untreated CP-affected subjects report “severe” or “very severe” pain. This lack of treatment for severe CP might point to an “unexpressed suffering” resulting from several factors, among which socioeconomic status and education appear to play a primary role. Additional contributing factors may include prior negative experience with pain therapy, culturally driven behaviors (e.g., stoic attitudes in the face of pain), or psychological elements such as poor treatment expectations, fear of bothering others, a resigned attitude, or significant concerns about opioids and other pain medications [7,62,63,64,65]. Undoubtedly, untreated pain leads to serious psychosocial and economic costs [66] and deserves attention from national healthcare systems.

Nevertheless, the proportion of CP-affected subjects who do not seek treatment recorded in our survey is far lower than that recently reported in other countries [67,68,69]. Indeed, the proportion of “untreated CP” may have been partially mitigated in Italy by a dedicated law on pain management issued in 2010 (Law n. 38, 15th March 2010). This law established, for the first time in the country, access to pain therapy as a fundamental right for all citizens and laid the foundation for the development of a nationwide network of inpatient and outpatient pain therapy centers. To date, besides Italy, pain management represents a legal obligation only in few European countries [70].

Finally, our survey highlighted that the effectiveness of CP treatments is perceived as partial or null by relevant proportions of people, 69% and nearly 8% of CP sufferers, respectively. Our survey did not identify which treatments were perceived as more or less effective, and this could be the focus of future investigations. Overall, these results underline the need for further investments in the pain therapy network and in developing biomedical and psychosocial research on the topic. In this framework, potential psychosocial and cultural factors underlying the phenomenon of “unexpressed/untreated” pain represent a critical issue warranting thorough investigation.

### 4.1. Policy and Clinical Implications

Our findings provide policymakers and healthcare professionals with up-to-date and accurate data to inform resource allocation, healthcare planning, and intervention strategies.

In Italy, a considerable proportion of individuals affected by CP experience only partial relief from treatments or undergo no treatment at all for severe or very severe pain. These individuals frequently present with co-occurring mental health problems, more than CP-free individuals. Considering the organizational model of the Italian National Health Service, characterized by regional fragmentation of care and limited integration of physical and mental health services [70,71], our results highlight the need for a wider implementation of integrated, multidisciplinary pain management programs. Such programs should include routine mental health screening in pain clinics, improved referral systems to mental health services or multidisciplinary centers, the development of tele-health services to promote accessibility, enhanced education for healthcare providers regarding the CP–mental health relationship, and public health campaigns to raise awareness on this relationship while reducing stigma toward psychological care.

### 4.2. Limitations and Strengths

The present analysis has inherent limitations typical of descriptive cross-sectional surveys, based on self-reported interviews. First, recall bias and self-reporting might have affected the accuracy of information collected on CP and psychosocial variables. Second, the cross-sectional design, with simultaneous assessment of exposures and outcomes, precludes any inference of causality. Third, the definition of a few variables presents further limitations: (i) no information about severity is recorded for depression and severe chronic anxiety; (ii) medication and other treatments are grouped together, limiting insights into specific therapeutic effects; and (iii) some variables are not pain-specific (e.g., functional limitations may or may not be pain-related). All of these limitations could have an impact on the internal validity of our study.

The absence of racial/ethnic data is another limitation, reflecting difficulties in estimating the size and impact of ethnic subgroups in Italy. Given the increasing immigrant population and improved demographic data collection, future surveys should consider including these variables to better understand CP disparities. Finally, the survey may have underestimated the prevalence of the most severe CP cases, particularly subjects with severe diseases with poor prognosis. These further limitations could affect the external validity of the study, restricting the generalizability of the results.

Nonetheless, it should be noted that self-reporting can be effective in very large population-based surveys. In this work, the self-report approach was a useful means to overcome the limits of clinical studies, where CP detection is often focused only on the most severe CP cases and does not capture information on the side of patients (e.g., patients’ perceptions regarding past or present treatment efficacy). Moreover, our study benefits from the strength of using a validated CP questionnaire administered to a large, representative sample of the Italian general population.

Finally, we may have contributed to overcoming the obstacles to the epidemiological comparability of CP across nations [3], by adopting the recent IASP CP definition, as several CP experts have done [6,11,72,73,74].

## 5. Conclusions

This work offers updated and essential insights into the prevalence and psychosocial burden of CP in Italy, establishing a crucial foundation for ongoing, systematic monitoring. Such continuous “surveillance” is important to ensure that public health policies and interventions remain evidence-based, adaptive to emerging trends, and effective in improving the quality of life for those living with CP. Since the data were collected in 2019, prior to the COVID-19 pandemic, they may also serve as a valuable baseline for assessing potential post-COVID-19 trends in CP prevalence. Indeed, a few studies already suggest that these trends are rising [75]. The EHIS will be replicated in 2025 and every six years thereafter. Since 2022, the CP-QUEST has also been included in another national survey, “Aspects of daily life”, which is expected to be conducted every two years and alternated with the EHIS, to ensure systematic information coverage on CP in Italy. Exploring the burden of CP in the general population will allow better identification of at-risk groups and support the development of targeted prevention and treatment initiatives. At the international level, these data may also help raise awareness of the magnitude of CP in different countries, thereby fostering broader public understanding of the problem.

## Figures and Tables

**Figure 1 ijerph-22-01395-f001:**
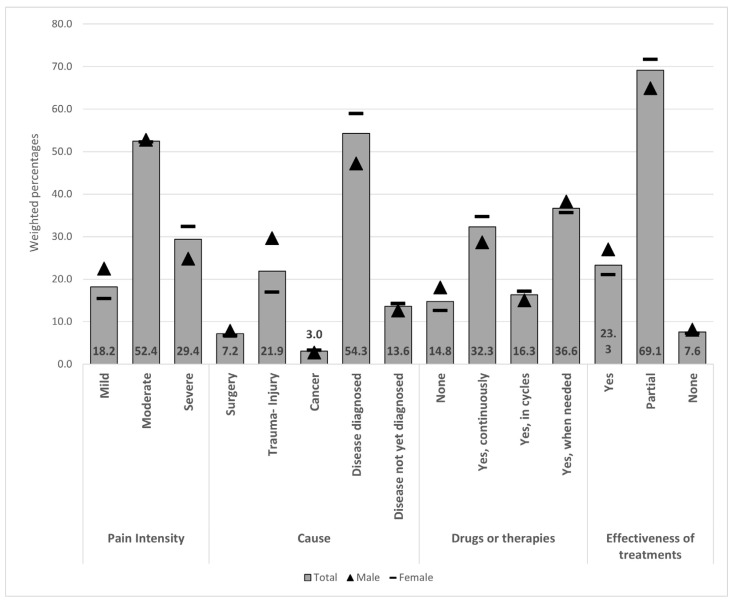
Chronic pain characteristics, causes, treatments, and self-perceived effectiveness by sex. Values at the bottom indicate total weighted percentages.

**Figure 2 ijerph-22-01395-f002:**
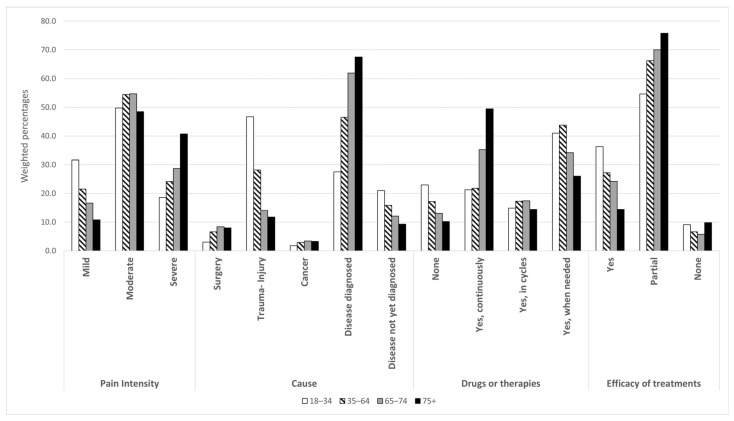
Chronic pain characteristics, causes, treatments, and self-perceived effectiveness by age classes.

**Figure 3 ijerph-22-01395-f003:**
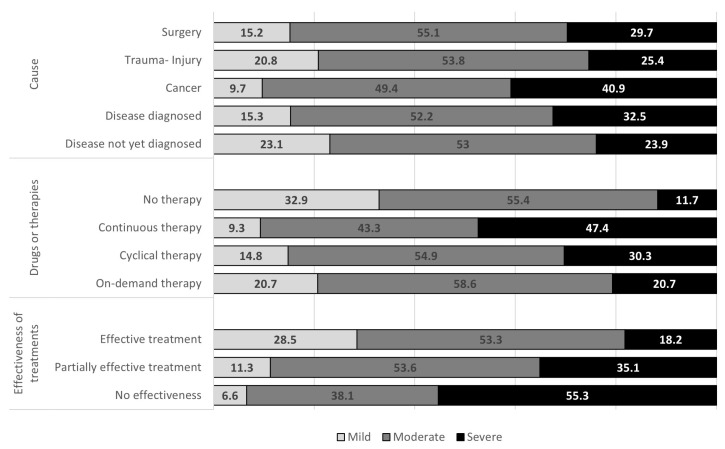
Chronic pain (CP) intensity levels by cause, therapy, and self-perceived effectiveness of treatments, per 100 respondents with CP.

**Table 1 ijerph-22-01395-t001:** Characteristics of EHIS participants with and without chronic pain (CP).

	No CP			CP		
	n = 28,825	% (75.9)	(95% CI)	n = 9950	% (24.1)	(95% CI)
**Sex**						
Male	14,459	50.8	(50.2–51.4)	3889	39.3	(38.2–40.5)
Female	14,366	49.2	(48.6–49.8)	6061	60.7	(59.6–61.8)
**Age class**						
18–34	6533	25.2	(24.7–25.8)	535	7.0	(6.4–7.6)
35–64	15,465	54.3	(53.7–54.9)	4268	44.9	(43.8–46.1)
65–74	3808	11.5	(11.1–11.9)	2088	19.5	(18.6–20.4)
75+	3019	9.0	(8.6–9.4)	3059	28.6	(27.6–29.7)
**Education level**						
High	5190	17.2	(16.7–17.6)	946	9.4	(8.7–10.1)
Medium	11,923	42.3	(41.7–42.9)	2911	29.7	(28.7–30.8)
Low	11,712	40.5	(39.9–41.1)	6093	60.9	(59.8–62.0)
**Limitations in activities due to health problems** (missing 97)						
Yes	3982	12.8	(12.4–13.2)	5959	58.3	(57.2–59.4)
No	24,759	87.2	(86.8–87.6)	3978	41.7	(40.6–42.8)
**Difficulties in social participation** (missing 21)						
Yes	3396	11.7	(11.3–12.1)	3241	32.3	(31.3–33.4)
No	25,417	88.3	(87.9–88.7)	6700	67.7	(66.6–68.7)
**Absence from work *** (missing 98)						
Yes	2584	18.0	(17.3–18.6)	1088	36.7	(34.7–38.7)
No	11,609	82.0	(81.4–82.7)	1884	63.3	(61.3–65.3)
**Depression** (missing 211)						
Yes	668	2.1	(1.9–2.3)	1210	11.3	(10.6–12.1)
No	27,982	97.9	(97.7–98.1)	8704	88.7	(88.0–89.4)
**Severe chronic anxiety** (missing 211)						
Yes	474	1.5	(1.3–1.6)	844	8.1	(7.5–8.8)
No	28,176	98.5	(98.4–98.7)	9070	91.9	(91.3–92.5)
**Primary care consultations** (missing 19)						
Yes	7151	23.6	(23.1–24.2)	5408	53.1	(52.0–54.3)
No	21,661	76.4	(75.8–76.9)	21,661	46.9	(45.7–48.0)
**Specialist medical consultations** (missing 623)						
Yes	4230	14.5	(14.1–15.0)	3219	33.1	(32.0–34.2)
No	24,193	85.5	(85.1–85.9)	6510	66.9	(65.8–68.0)
**Diagnostic procedures** (missing 358)						
Yes	10,470	35.4	(34.8–36.0)	6128	62.2	(61.1–63.3)
No	18,077	64.6	(64.0–65.2)	3742	37.8	(36.7–38.9)
**Multimorbidity** (missing 197)						
Yes	4598	14.7	(14.2–15.1)	5956	58.6	(57.4–59.6)
No	24,058	85.3	(84.9–85.7)	3966	41.4	(40.3–42.5)

Sample size (n) and weighted percentages (%). All comparisons, no CP vs. CP, were significant at *p*-value < 0.001. * Absence from work due to health problems, only for employed and self-employed respondents.

**Table 2 ijerph-22-01395-t002:** Chronic pain in Italy by age and sex.

	Male			Female			All		
Age Class	n	%	(95% CI)	n	%	(95% CI)	n	%	(95% CI)
18–34	263	8.0	(7.0–9.0)	272	8.0	(7.0–9.0)	535	8.0	(7.3–8.7)
35–64	1794	17.9	(17.0–18.7)	2474	23.7	(22.8–24.6)	4268	20.8	(20.2–21.4)
65–74	816	28.6	(26.5–30.7)	1272	41.0	(38.8–43.1)	2088	35.1	(33.7–36.6)
75+	1016	40.9	(38.4–43.3)	2043	56.7	(54.7–58.7)	3059	50.3	(48.8–51.9)
**Total**	**3889**	**19.7**	**(19.1–20.4)**	**6061**	**28.1**	**(27.5–28.8)**	**9950**	**24.1**	**(23.6–24.6)**

Sample size (n) and weighted percentages (%).

**Table 3 ijerph-22-01395-t003:** Social outcomes and chronic pain (CP) associations.

Outcome Variables												
	Limitations in Activities Due to Health Problems				Difficulties in Social Participation				Absence from Work *			
	%	(95% CI)	ORadj	(95% CI)	%	(95% CI)	ORadj	(95% CI)	%	(95% CI)	ORadj	(95% CI)
**Independent variable**												
**Chronic pain**												
No CP	12.8	(12.4–13.2)	1		11.7	(11.3–12.1)	1		18.0	(17.3–18.6)	1	
Mild CP	34.6	(31.9–37.3)	3.02	(2.69–3.40)	19.4	(17.1–21.7)	1.65	(1.45–1.87)	27.6	(23.6–31.6)	1.72	(1.45–2.03)
Moderate CP	55.7	(54.1–57.3)	6.25	(5.81–6.72)	28.9	(27.4–30.4)	2.54	(2.35–2.74)	36.3	(33.5–39.1)	2.48	(2.22–2.77)
Severe CP	78.1	(76.3–79.9)	15.99	(14.40–17.76)	46.2	(44.1–48.4)	4.82	(4.41–5.27)	49.2	(44.2–54.1)	4.18	(3.53–4.94)
**Covariates**												
**Sex**												
Male	20.7	(20.0–21.3)	1		14.1	(13.4–14.5)	1		19.2	(18.3–20.0)	1	
Female	26.6	(25.8–27.1)	1.03	(0.96–1.10)	19.2	(18.5–19.7)	1.23	(1.16–1.30)	23.7	(22.5–24.7)	1.24	(1.15–1.33)
**Age class**												
18–34	5.7	(5.1–6.3)	1		11.1	(10.3–11.9)	1		16.8	(15.4–18.1)	1	
35–64	16.4	(15.7–16.9)	2.20	(1.97–2.44)	13.8	(13.2–14.3)	1.07	(0.99–1.16)	22.3	(21.5–23.1)	1.28	(1.16–1.41)
65–74	39.2	(37.6–40.6)	5.84	(5.20–6.58)	15.8	(14.6–16.9)	1.03	(0.92–1.15)	20.4	(15.4–25.3)	1.06	(0.84–1.41)
75+	64.0	(62.4–65.3)	13.23	(11.73–14.92)	37.0	(35.3–38.3)	2.74	(2.48–3.03)	26.5	(9.2–43.8)	1.23	(0.62–2.43)
**Education level**												
High	12.2	(11.1–13.2)	1		15.9	(15.5–16.2)	1		20.8	(19.5–21.9)	1.00	
Medium	15.1	(14.4–15.7)	1.17	(1.06–1.30)	12.9	(12.3–13.5)	0.74	(0.68–0.81)	20.6	(19.6–21.5)	0.98	(0.89–1.08)
Low	35.1	(34.2–35.8)	1.71	(1.55–1.89)	20.3	(18.7–21.7)	0.81	(0.74–0.88)	22.1	(20.4–23.7)	1.01	(0.91–1.13)
**AUC**	0.84				0.70				0.61			

* Absence from work due to health problems, only for employed and self-employed respondents.

**Table 4 ijerph-22-01395-t004:** Depression and severe chronic anxiety and chronic pain (CP) associations.

	Outcome Variables							
	Depression				Severe Chronic Anxiety			
	%	(95% CI)	ORadj	(95% CI)	%	(95% CI)	ORadj	(95% CI)
**Independent variable**								
**Chronic pain**								
No CP	2.1	(1.9–2.3)	1		1.5	(1.3–1.6)	1	
Mild CP	3.6	(2.5–4.8)	1.42	(1.08–1.86)	3.0	(1.9–4.0)	1.71	(1.26–2.30)
Moderate CP	8.9	(8.0–9.9)	3.09	(2.70–3.53)	6.2	(5.4–7.1)	3.09	(2.64–3.62)
Severe CP	20.2	(18.5–22.1)	7.10	(6.22–8.11)	14.7	(13.1–16.3)	7.03	(6.02–8.21)
**Covariates**								
**Sex**								
Male	2.7	(2.4–2.9)	1		1.8	(1.5–2.0)	1	
Female	5.9	(5.5–6.3)	1.78	(1.60–1.99)	4.3	(4.0–4.6)	1.97	(1.72–2.25)
**Age class**								
18–34	1.1	(0.8–1.3)	1		0.9	(0.7–1.2)	1	
35–64	3.1	(2.8–3.4)	2.12	(1.68–2.68)	2.3	(2.0–2.5)	1.72	(1.34–2.23)
65–74	7.4	(6.5–8.2)	3.74	(2.92–4.81)	4.8	(4.1–5.5)	2.52	(1.90–3.33)
75+	11.3	(10.0–12.0)	3.90	(3.03–5.01)	7.8	(6.9–8.7)	2.87	(2.18–3.78)
**Education level**								
High	2.1	(1.7–2.6)	1		4.7	(4.4–5.1)	1	
Medium	2.6	(2.3–2.9)	1.16	(0.94–1.43)	1.8	(1.5–2.0)	1.10	(0.86–1.40)
Low	6.6	(6.2–7.0)	1.62	(1.33–1.97)	1.5	(1.1–1.9)	1.68	(1.33–2.12)
**AUC**	0.79				0.78			

**Table 5 ijerph-22-01395-t005:** Primary and specialist care consultations, diagnostic procedures, and chronic pain (CP) associations.

	Outcome Variables											
	Primary Care Consultations				Specialist Medical Consultations				Diagnostic Procedures			
	%	(95% CI)	ORadj	(95% CI)	%	(95% CI)	ORadj	(95% CI)	%	(95% CI)	ORadj	(95% CI)
**Independent variables**												
**Chronic pain**												
No CP	23.6	(23.1–24.2)	1		14.5	(14.1–15.0)	1		35.4	(34.8–36.0)	1	
Mild CP	38.6	(35.8–41.3)	1.51	(1.36–1.69)	24.3	(21.8–26.8)	1.56	(1.38–1.76)	56.5	(53.7–59.3)	2.02	(1.82–2.24)
Moderate CP	50.6	(49.0–52.2)	1.89	(1.76–2.02)	31.2	(29.7–32.7)	1.88	(1.74–2.03)	62.6	(61.1–64.2)	2.24	(2.09–2.40)
Severe CP	66.2	(64.1–68.3)	2.90	(2.64–3.18)	42.1	(39.9–44.3)	2.63	(2.40–2.89)	65.3	(63.3–67.4)	2.27	(2.07–2.49)
**Covariates**												
**Sex**												
Male	26.4	(25.7–27.1)	1		15.6	(15.0–16.2)	1		33.8	(33.0–34.5)	1	
Female	34.6	(33.9–35.3)	1.26	(1.20–1.32)	22.0	(21.4–22.7)	1.32	(1.25–1.39)	49.8	(49.0–50.6)	1.80	(1.72–1.88)
**Age class**												
18–34	16.0	(15.1–16.9)	1		13.2	(12.3–14.1)	1		27.9	(26.8–29.0)	1	
35–64	25.9	(25.2–26.5)	1.40	(1.31–1.50)	16.6	(16.0–17.2)	1.08	(1.00–1.17)	43.4	(42.6–44.1)	1.78	(1.67–1.88)
65–74	44.5	(43.0–46.0)	2.33	(2.14–2.55)	24.8	(23.4–26.1)	1.41	(1.28–1.56)	51.6	(50.1–53.2)	2.05	(1.89–2.23)
75+	57.9	(56.4–59.4)	3.11	(2.84–3.45)	30.9	(29.5–32.4)	1.58	(1.42–1.75)	49.6	(48.1–51.1)	1.52	(1.39–1.66)
**Education level**												
High	23.4	(22.2–24.7)	1		19.9	(18.7–21.1)	1		46.3	(44.9–47.8)	1	
Medium	24.8	(24.1–25.6)	1.05	(0.97–1.13)	17.1	(16.4–17.8)	0.79	(0.73–0.85)	41.6	(40.7–42.4)	0.81	(0.76–0.86)
Low	38.2	(37.4–39.0)	1.14	(1.06–1.22)	20.2	(19.5–20.9)	0.63	(0.58–0.68)	41.2	(40.4–42.0)	0.54	(0.50–0.58)
**Multimorbidity**												
No	22.3	(21.7–22.8)	1		14.1	(13.7–14.6)	1		35.6	(35.1–36.2)	1	
Yes	56.0	(54.9–57.0)	2.20	(2.08–2.33)	33.2	(32.1–34.2)	1.95	(1.82–2.08)	61.1	(60.0–62.2)	2.04	(1.93–2.16)
**AUC**	0.73				0.68				0.69			

**Table 6 ijerph-22-01395-t006:** Socio-demographic characteristics of the EHIS respondent subgroup with a self-reported diagnosis of depression or severe chronic anxiety (with and without chronic pain, CP).

	No CP			CP			All		
	n = 911	% (39.0)	(95% CI)	n = 1500	% (61.0)	(95% CI)	n = 2411	% (100)	(95% CI)
**Sex**									
Male	320	35.8	(31.9–39.7)	383	25.3	(22.5–28.2)	703	29.4	(27.1–31.7)
Female	591	64.2	(60.3–68.1)	1117	74.7	(71.9–77.4)	1708	70.6	(68.4–72.8)
**Age class**									
18–34	73	10.1	(7.5–12.6)	36	3.0	(1.8–4.2)	109	5.8	(4.6–7.0)
35–64	418	46.5	(42.4–50.6)	478	33.2	(30.1–36.2)	896	38.4	(36.0–40.8)
65–74	207	21.8	(18.4–25.3)	358	23.0	(20.2–25.7)	565	22.5	(20.4–24.6)
75+	213	21.6	(18.2–25.0)	628	40.8	(37.7–40.0)	841	33.3	(31.0–35.7)
**Education level**									
High	108	11.5	(8.8–14.2)	88	5.8	(4.2–7.4)	196	8.0	(6.6–9.4)
Medium	255	28.0	(24.3–31.8)	315	20.9	(18.3–23.6)	570	23.7	(21.6–25.8)
Low	548	60.5	(56.5–64.4)	1097	73.3	(70.5–76.1)	1645	68.3	(66.0–70.5)
**Consultation with a psychiatrist, psychologist, or psychotherapist** (missing 2)									
Yes	306	33.9	(32.3–35.5)	330	22.2	(21.0–23.3)	636	25.8	(25.8–27.7)
No	605	66.4	(64.4–67.6)	1168	77.8	(76.6–78.9)	1773	72.3	(72.2–74.1)
**Foregone mental health care due to financial barriers** (missing 14)									
Yes	42	5.2	(3.3–7.2)	105	6.9	(5.2–8.6)	147	6.3	(5.0–7.5)
No	531	59.3	(55.3–63.3)	797	53.6	(50.4–56.8)	1328	55.8	(53.4–58.3)
No need for mental health care	332	35.5	(31.6–39.4)	590	39.5	(36.3–42.6)	922	37.9	(35.5–40.3)

Sample size (n) and weighted percentages (%).

**Table 7 ijerph-22-01395-t007:** Mental health care access and needs in respondents with a self-reported diagnosis of depression and/or severe chronic anxiety, as well as chronic pain (CP) associations.

	Outcome Variables											
	Consultation with a Psychiatrist, Psychologist, or Psychotherapist				Foregone Mental Health Care due to Financial Barriers				No Need for Mental Health Care			
	%	(95% CI)	ORadj	(95% CI)	%	(95% CI)	ORadj	(95% CI)	%	(95% CI)	ORadj	(95% CI)
**Independent variables**												
**Chronic pain**												
No	33.9	(32.3–35.5)	1		5.2	(3.3–7.2)	1		35.5	(31.6–39.4)	1	
Yes	22.2	(21.0–23.3)	0.75	(0.61–0.92)	6.9	(5.2–8.6)	1.57	(1.07–2.31)	39.5	(36.3–42.6)	1.02	(0.85–1.23)
**Covariates**												
**Sex**												
Male	32.5	(28.0–36.9)	1		5.0	(2.8–7.2)	1		33.6	(29.0–38.1)	1	
Female	24.4	(21.8–27.0)	0.83	(0.70–1.03)	6.8	(5.2–8.3)	1.51	(0.99–2.30)	39.7	(36.8–42.7)	1.20	(0.98–1.46)
**Age class**												
18–34	63.1	(52.2–74.0)	1		9.4	(2.5–16.4)	1		21.0	(11.6–30.4)	1	
35–64	35.9	(31.9–39.9)	0.37	(0.25–0.54)	8.9	(6.5–11.4)	0.76	(0.39–1.47)	33.5	(29.6–37.4)	1.79	(1.13–2.83)
65–74	20.7	(16.2–25.3)	0.18	(0.12–0.28)	3.9	(1.6–6.2)	0.29	(0.13–0.63)	40.5	(35.1–45.9)	2.35	(1.46–3.79)
75+	14.0	(10.8–17.2)	0.12	(0.08–0.19)	4.2	(2.2–6.1)	0.28	(0.13–0.60)	44.4	(39.9–48.8)	2.72	(1.69–4.36)
**Education level**												
High	35.8	(26.6–45.0)	1		5.0	(0.6–9.4)	1		30.8	(21.9–39.7)	1	
Medium	35.0	(29.8–40.1)	1.10	(0.74–1.59)	7.0	(4.2–9.9)	1.41	(0.65–3.05)	35	(29.8–40.1)	1.15	(0.79–1.68)
Low	22.9	(20.3–25.5)	0.89	(0.62–1.27)	6.1	(4.6–7.7)	1.54	(0.73–3.22)	39.8	(36.8–42.8)	1.20	(0.85–1.71)
**AUC**		0.69				0.67					0.5	

## Data Availability

The whole datasets generated and/or analyzed during the current work are not publicly available due to the stewardship of personal data by ISTAT (Italian Bureau of Statistics); data queries are available from the corresponding author upon reasonable request, provided that the request meets the criteria of ISTAT for data release.

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
