# Peer review of "Chronic Pain Prevalence and Psychosocial Burden in the Italian Population from the 2019 European Health Interview Survey"

_ijerph, 2025, doi:10.3390/ijerph22091395_

Round 1
Reviewer 1 Report
Comments and Suggestions for Authors
The manuscript addresses a timely and relevant study on CP and it’s burden. A large dataset is used as well as a validated questionnaire.
Abstract:
The Abstract does not include all the information and I feel line 16 – 30 is confusing. Does the results refer to the current study or the 2019 EHIS? It is unclear of this is a new study or using the 2019 data, just eliminating the below 18 years of age data. If not, the current methodology and sampling strategy should be declared.
Introduction:
It seems in the lit review that the 2019 data is being used, but claims to “update” CP prevalence estimates. If the CP prevalence is to be updated with old data, it needs to be explained how, here.
The methodology section is unclear in that it refers to the 2019 sample or the current study’s sample. I do not think the methodology section is suppose to refer to the 2019 data collection, but rather how the current (secondary) data was obtained. The personal data (207) is also confusing as secondary data was used. The data analysis refers to the original data. Line 269 reports ont eh small sample size, but I cannot see the sample size of the sub-population, although the study might become clearer now that the 2019 data was used with different inclusion and exclusion criteria?
The Result section is well written. Good use was made using graphs and tables.
The Discussion (406) again talks about the “current” study. Should it not be “current” analysis?
The manuscript is well-researched, clearly written, and presents valuable findings relevant to public health. The strengths significantly outweigh the weaknesses, and the authors have appropriately acknowledged the limitations of their study design. It would be nice if the Abstract and Methodology sections can be clarified. I also suggest integration of results and Lit Review in the Discussion section.
Specific issues:
- The Abstract needs to reflect the current study better. It does not address the questions: how the data was obtained (for the current study?), and the findings of the current study, as well as the significance of the study.
- Line 54 – 77 repetitive.
- It is unclear whether this is the “first” article on the data obtained in 2019. Was the “analytical team” only constituted in 2022? (line 87)
- Methodology section needs to be rewritten.
- Line 127 – citation style
- Line 269 – sample size.
- Minor typing/editing/technical errors. (eg., 525)
- Language editing
I feel that a language editor might assist in making the manuscript clearer.
Reviewer 2 Report
Comments and Suggestions for Authors
This is a timely and important study that provides updated national estimates on the prevalence and psychosocial impact of chronic pain in Italy, using high-quality EHIS data and a validated assessment tool. The methodology is sound and the findings are clearly presented. That said, the manuscript would benefit from more concise interpretation of results, a clearer discussion around treatment patterns, and stronger comparisons with similar international studies. Addressing potential bias from survey non-respondents, refining some terminology, and making minor language edits would further enhance clarity. Overall, this is a well-executed study that deserves publication after minor revisions.
To enhance the clarity and impact of the manuscript, the authors could consider streamlining dense tables—especially those stratified by age and sex—and improving visual elements such as figures for easier interpretation. Further elaboration on gender differences in CP, including possible biological and social factors, would add depth to the discussion. The surprising finding of reduced mental health service use in those with both CP and psychiatric diagnoses deserves additional interpretation, possibly exploring barriers such as stigma or physical disability. Expanding on the policy and clinical implications—such as how these data might guide multidisciplinary pain care or inform reimbursement models—would also strengthen the manuscript’s relevance. Lastly, the limitations section could be broadened to include potential biases related to self-reporting and the cross-sectional design. Overall, with a few refinements, this is a valuable and timely contribution to chronic pain epidemiology.
Reviewer 3 Report
Comments and Suggestions for Authors
Please see the file.

Round 2
Reviewer 1 Report
Comments and Suggestions for Authors
I would like to thank the authors for their positive attitude towards the review and for clarification of my comments. This study is valuable and makes a contribution to academia.